# Measles and Rubella Elimination in the Western Pacific Region in 2013–2022: Lessons Learned from Progress and Achievements Made during Regional and Global Measles Resurgences

**DOI:** 10.3390/vaccines12070817

**Published:** 2024-07-22

**Authors:** Yoshihiro Takashima, Syeda Kanwal Aslam, Roger Evans, Kayla Mae Mariano, Chung-won Lee, Xiaojun Wang, Varja Grabovac, David N. Durrheim

**Affiliations:** 1World Health Organization Regional Office for the Western Pacific, Manila 1000, Philippines; saslam@who.int (S.K.A.); revans@who.int (R.E.); marianok@who.int (K.M.M.); wangxia@who.int (X.W.); grabovacv@who.int (V.G.); 2School of Medicine and Public Health, University of Newcastle, Callaghan, NSW 2308, Australia; david.durrheim@health.nsw.gov.au

**Keywords:** measles elimination, rubella elimination, sustainability of elimination, regional resurgence of measles, global resurgence of measles, import-related outbreak, global synchronization of elimination effort

## Abstract

Measles is the most contagious communicable disease, causing an estimated 5.5 million cases and more than 30,000 deaths in the Western Pacific Region (WPR) during 2000. Rubella infection in a pregnant woman can be devastating for the foetus, resulting in congenital rubella syndrome (CRS) in 90% of rubella infections in early pregnancy. It was estimated that approximately 9000 CRS cases occurred in the WPR in 2010. World Health Organization (WHO) Member States in the WPR decided in 2003 to eliminate measles and in 2014 to eliminate rubella from the region. While the WPR successfully attained historically low measles incidence in 2012, it experienced a region-wide measles resurgence in 2013–2016. During the regional resurgence, WHO and Member States accumulated greater knowledge on the epidemiology of measles and rubella in the WPR and strategies to maintain gains. The implementation of the resulting new regional strategy and plan of action from 2018 has proven that measles and rubella elimination is achievable and sustainable under the pressure of multiple importations of measles virus during the world-wide measles resurgences in 2018–2019. This article discusses this progress and achievements towards achieving the global eradication of measles and rubella.

## 1. Background

Measles is a highly contagious and serious disease caused by the measles virus. It was estimated that 5.5 million measles cases and 31,200 to 34,000 deaths occurred in 2000 in the Western Pacific Region (WPR) [1,2]. While rubella is usually a mild viral disease in children, infection in a pregnant woman can be devastating for the foetus. Congenital rubella syndrome (CRS) occurs in 90% of maternal rubella infections in early pregnancy. The estimated CRS incidence was 11,541 cases in 1996, 11,084 in 2000 and 8889 in 2010 in the Western Pacific Region [3].

In 2003, the World Health Organization (WHO) Regional Committee (RC) for the Western Pacific confirmed that measles elimination should be a regional goal, endorsed the Western Pacific Regional Plan of Action for Measles Elimination and urged Member States to offer all children two doses of measles-containing vaccine (MCV) to achieve and maintain a 95% population immunity of each birth cohort in every district and use measles elimination to strengthen routine immunization and other public health programmes, such as the prevention of CRS (WPR/RC54.R3) [4]. In 2005, the RC decided that the WPR should aim to eliminate measles by 2012 and urged Member States to develop or strengthen national plans for measles elimination as part of comprehensive multi-year plans for immunization services (WPR/RC56.R8) [5].

In 2010, the RC reaffirmed the 2012 measles elimination goal, requested WHO to establish regional verification mechanisms for measles elimination and urged Member States to commit the human and financial resources necessary to achieve and sustain measles elimination, accelerate the control of rubella and the prevention of CRS and establish an independent verification process for measles elimination (WPR/RC61.R7) [6].

Countries and areas in the WPR have progressively strengthened their national immunization programmes and significantly reduced measles transmission, morbidity and mortality. In 2012, the region recorded the lowest number of measles cases in its history—10,794 cases, or 5.9 cases per 1 million population, compared with 177,052 cases, or 105.1 cases per 1 million population, in 2000 (Figure 1).

In 2012, the RC reaffirmed its commitment to eliminate measles and accelerate rubella control in the WPR, and urged Member States to implement effective immunization strategies to identify and reach all vulnerable underserved communities in both rural and urban settings, interrupt all residual endemic measles virus transmission as rapidly as possible and establish national verification committees (NVCs) to develop regular progress reports for submission to the Regional Verification Commission for Measles Elimination in the Western Pacific (RVC) (WPR/RC63.R5) [8]. In 2014, the RC endorsed a regional rubella elimination goal as one of eight regional immunization goals in the Regional Framework for Implementation of the Global Vaccine Action Plan in the Western Pacific (WPR/RC65.R5) [9].

## 2. Regional Measles Resurgence in the Western Pacific 2013–2016

After the low number of measles cases in 2012, the Western Pacific experienced a region-wide measles resurgence from 2013 to 2016. The regional measles incidence rate per 1 million population increased from 5.9 in 2012 to 19.4 in 2013, 69.2 in 2014, 36.2 in 2015 and 30.2 in 2016 (Figure 1).

### 2.1. Resurgence of Measles Virus Transmission in Endemic Countries

There were large national resurgences of measles virus transmission in endemic countries. In China, the ongoing transmission of the genotype H1 measles virus, which has been the only endemic genotype detected in China since surveillance began in 1993 [10], significantly decreased as a result of multiple province-wide measles supplementary immunization activities (SIAs) between 2003 and 2009 and a nation-wide measles SIA in 2010, resulting in the lowest measles incidence ever reported in China in 2012. However, transmission then increased and there was a nation-wide measles resurgence in China which lasted until 2016 [11]. In the Philippines, the measles genotype D9 was endemic and caused two large outbreaks in 2010 and 2011, but the genotype B3 virus became endemic in early 2013 [12,13]. A nation-wide measles rubella-containing vaccine SIA (MRCV-SIA) rolled out in September 2014 had sub-optimal coverage with measles virus transmission continuing, particularly in the middle and southern parts of the Philippines (Visayas and Mindanao) [14], and the nation-wide measles resurgence lasted until 2015. Malaysia, where the genotype D8 measles virus has continued to be detected since 2013 and the genotype D9 virus was identified in 2015–2017, experienced increased measles virus transmission between 2015 and 2018.

### 2.2. Importation of Measles Virus from Countries with Increased Endemic Transmission or Import-Related Large-Scale Outbreaks

The multiple importations of the measles virus from countries with resurgences of endemic transmission or import-related large-scale outbreaks led to multiple small-scale outbreaks between 2013 and 2015 in Australia, Hong Kong SAR (China), Japan, the Republic of Korea, Singapore and Vanuatu [15,16,17].

### 2.3. Nation-Wide Measles Outbreaks after Importation

Several countries experienced large-scale outbreaks following importation after a period of low or no documented transmission: New Zealand (genotype B3 virus in 2013–2014), Viet Nam (genotype H1, B3 and D8 viruses in 2013–2014), Papua New Guinea (genotype D8 virus imported from Indonesia in 2013 and genotype B3 virus in 2014) [18], the Federated States of Micronesia (genotype B3 virus in 2014), Solomon Islands (genotype B3 virus imported from Papua New Guinea in 2014) [19], Lao PDR (genotype H1 virus in 2014) and Mongolia (genotype H1 virus in 2015–2016) [20]. These importations resulted in large-scale measles outbreaks in 2013–2016, revealing important population immunity gaps [20].

### 2.4. Re-Established Measles Virus Transmission after Elimination

In Mongolia, which was verified by the RVC in March 2014 to have achieved and sustained the interruption of endemic measles virus transmission for more than three years, a prolonged nation-wide measles outbreak began in March 2015 due to the imported genotype H1 virus [20], leading to classification by the RVC as having re-established endemic transmission as of September 2017 [21].

### 2.5. Response to the Regional Resurgence

In response to the regional resurgence, the Technical Advisory Group (TAG) for Immunization and Vaccine-Preventable Diseases in the Western Pacific Region recommended WHO develop a new strategy and plan of action for measles elimination in the WPR [22], which was submitted to and endorsed by the RC in 2017 as the Regional Strategy and Plan of Action for Measles and Rubella Elimination in the Western Pacific (WPR/RC68.R1) [23].

The Regional Strategy and Plan of Action proposed Operational Targets for 2020 as milestones to attain the regional goal, which was not only to achieve but also sustain the elimination of measles and rubella in all countries and areas of the WPR. The targets included: (i) sustain the interruption of measles virus transmission in countries and areas that have reached measles elimination; (ii) prevent the resurgence of endemic measles virus transmission (genotypes B3, D8, D9 and H1); (iii) interrupt all ongoing measles transmission in endemic countries; and (iv) prevent large-scale outbreaks after importation for measles elimination; (v) set a national target year for rubella elimination in all countries; and (vi) establish CRS surveillance in all countries for rubella elimination [24].

## 3. Progress in Achieving and Sustaining Measles and Rubella Elimination in the Western Pacific Region after the 2013–2016 Regional Resurgence

### 3.1. Elimination of Measles and Rubella Verified by RVC from 2014 to 2023

The Regional Verification Commission (RVC) was established in January 2012. In line with the resolution WPR/RC63.R5 in 2012, all 14 countries and 2 areas in the Western Pacific Region established National Verification Committees (NVCs) and the Pacific Island countries and areas (PICs), consisting of 21 countries and areas in the Pacific (which are considered one epidemiological block for the purpose of the verification of measles and rubella elimination), established a Sub-Regional Verification Committee (SRVC). NVCs and SRVC started the submission of annual progress reports towards measles and rubella elimination in the second half of 2013 and an annual meeting of the RVC has occurred since May 2014 to monitor progress towards elimination attainment.

The verification of measles and rubella elimination in the WPR requires the documentation of the interruption of endemic measles and rubella virus transmission for a period of at least 36 months from the last known endemic case [25]. From 2014 to 2023, eight countries and two areas in the region were verified to have achieved measles elimination and, among them, six countries and two areas were verified in 2023 to have sustained measles elimination. Since 2017, a total of five countries and two areas in the region were verified to have achieved and sustained rubella elimination (Figure 2).

Figure 1 shows the status of measles and rubella elimination in countries, areas and the Pacific sub-region from 2014 to 2023 by (i) the status of measles elimination in 2023; (ii) sustainability of measles elimination; and (iii) the size of total population.

### 3.2. Measles Incidence

While Singapore, New Zealand, Macao SAR (China), Cambodia, Lao PDR, Samoa, Tonga and American Samoa experienced significant increases in measles incidence during the global measles resurgence in 2018 to 2019 [26,27,28], all six countries and two areas that had been verified by RVC to have achieved measles elimination, China, Papua New Guinea, Lao PDR and the Pacific sub-region have continued to achieve decrease in measles incidence (i.e., confirmed measles cases per million population) since 2019 and many of them have maintained a measles incidence of <1 per million population from 2021 to 2023 (Figure 2).

**Scheme 2 vaccines-12-00817-sch002:**
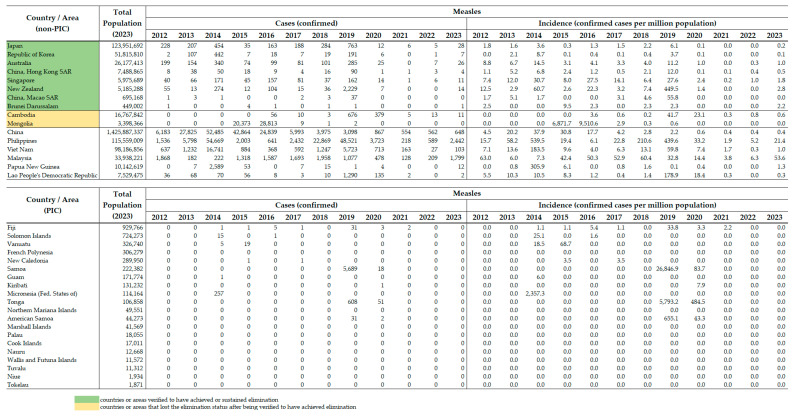
Confirmed cases and incidence of measles in the Western Pacific Region by country, 2012–2023. Source: [29,30].

### 3.3. Routine Immunization Programmes and Supplementary Immunization Activities (SIA)

All countries and areas which were verified by the RVC to have achieved and sustained measles elimination maintained >90% annual coverage of measles and rubella containing vaccine first dose (MRCV-1) from 2013 to 2022, and most countries maintained a >90% coverage of MRCV-2 from 2013 to 2022 (except Macao SAR (China) in 2013 (89.2%), New Zealand from 2013 to 2022 (82.3–90.7%) and Singapore in 2016 (87.9%) and 2018 (84.0%)) (Figure 3). Of these countries, only New Zealand experienced notable declines in MRCV-2 coverage during the COVID-19 pandemic in 2021–2022.

Among countries which have not yet been verified to have achieved measles elimination, China and Malaysia sustained > 90% annual coverage with both MRCV-1 and MRCV-2 in each year from 2013 to 2022 (except MRCV-2 coverage in 2019 (87.0%) and 2020 (84.0%) in Malaysia). Viet Nam, which reported a >90% coverage of both MRCV-1 and MRCV-2 in each year from 2014 to 2020, experienced a marked decline in coverage with both MRCV-1 and MRCV-2 during the COVID-19 pandemic response in 2021 and 2022. The routine coverage of both MRCV-1 and MRCV-2 never reached 80% in the Philippines and Papua New Guinea from 2013 to 2022, and the Philippines had decreasing coverage for both MRCV-1 and MRCV-2 during the COVID-19 pandemic in 2021 and 2022.

**Scheme 3 vaccines-12-00817-sch003:**
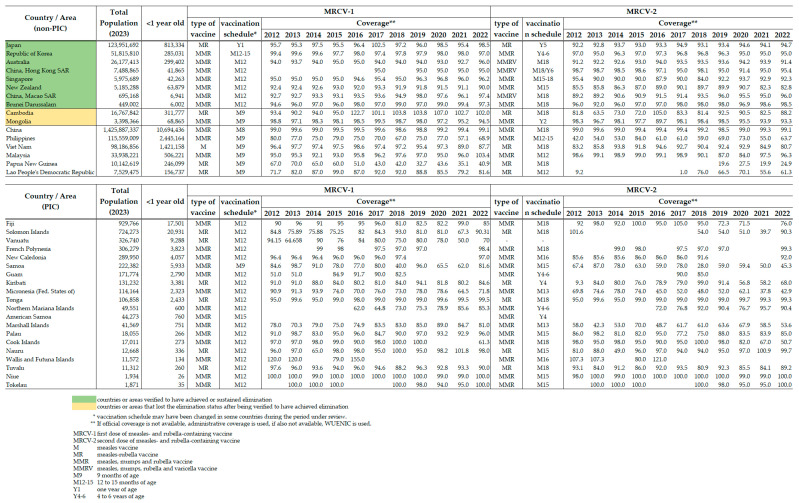
Coverage of MRCV-1 and MRCV-2 (routine immunization with 1st and 2nd doses of measles- and rubella-containing vaccines) in the Western Pacific Region by country, 2012–2022. Source: [31].

Japan, the Republic of Korea, Australia, Hong Kong SAR (China), Singapore, New Zealand, Macao SAR (China) and Brunei Darussalam were verified by the RVC to have achieved and then verified to have sustained measles elimination without an implementation of nation-wide MCV-SIAs between 2013 and 2023.

Mongolia conducted a nation-wide MCV-SIA, targeting children aged 6 months to 6 years, in May and June 2015, and another nation-wide MCV-SIA, targeting young adults aged 18 to 30 years, in May 2016, in response to the largest measles outbreak since the introduction of MCV into the national immunization programme in 1973 (Figure 4) [32]. This import-related outbreak in Mongolia predominantly affected young people aged 15 to 29 years followed by children aged 0 to 8 months (accounting for 59% and 12% of cases, respectively). Cambodia conducted a nation-wide MR-SIA targeting children aged 6 to 59 months in 2017 in response to an import-related measles outbreak in 2016.

The Philippines carried out nation-wide MR-SIAs targeting children aged 6 to 59 months in 2014 and 2020. Both SIAs followed periods of major resurgence in endemic transmission in 2013–2014 and 2018–2019. Fiji, Samoa and Tonga carried out nation-wide MR-SIAs targeting adults as well as children and adolescents during the latter periods of large-scale import-related measles outbreaks in 2019 in response to the global measles resurgence and importations from New Zealand.

While most MRCV-SIAs carried out in the WPR attained high vaccination coverage (>90%), their impact could have been substantially greater if they had targeted the specific groups with large immunity gaps and if they had been implemented before or very soon after the detection of resurgences or import-related outbreaks.

### 3.4. Surveillance

Several countries or areas verified by the RVC as having achieved measles elimination (e.g., Macao SAR (China), Brunei Darussalam, Cambodia and Mongolia) and some countries which have not yet achieved measles elimination (e.g., China, Malaysia, Lao PDR and the Pacific sub-region) have maintained highly sensitive measles and rubella surveillance measured by “discarded non-measles rate per 100,000 population” from 2013 to 2023 (Figure 5).

The Republic of Korea, Hong Kong SAR (China), Macao SAR (China), Brunei Darussalam, Cambodia and Mongolia (countries or areas verified by RVC to have achieved measles elimination) and China, Malaysia, Papua New Guinea and the Pacific sub-region (countries or sub-region which have not yet achieved measles elimination) have improved or sustained high case investigation performance, measured by “suspected cases with adequate investigation” (>80%), and sample collection, as reflected by “suspected cases with adequate specimens for laboratory confirmation” (>80%), from 2018 to 2023.

### 3.5. Genotypes of Measles Virus Detected in the Western Pacific Region

Genotype D9, which was considered endemic in Malaysia until the end of 2017 and detected frequently as imported by Japan, Australia, Hong Kong SAR (China) and China from 2013 to the middle of 2015 and by Singapore from 2013 to late 2017, has not been detected and reported in the region since January 2018 (Figure 6).

Genotype H1 has been the only endemic genotype detected in China since surveillance began in 1993 and was widely distributed throughout China from 1993 to 2010 [10,11]. It became endemic in Viet Nam in 2013–2015 and in Mongolia from 2015 to 2016 due to importation and was detected frequently as imported in Japan, the Republic of Korea, Australia, Hong Kong SAR (China), Macao SAR (China) and Lao PDR between 2013 and 2016. It has not been detected or reported in the WPR and globally since September 2019. Globally, the number of measles cases of genotype H1 declined from 2625 in 2016 to 333 in 2018 (87% decline), 28 in 2019 (99% decline) and 0 from 2020 onwards.

Genotype B3, which has been endemic in the Philippines since 2013 [14], continued to be detected as imported in all countries and areas of the region verified to have achieved measles elimination, except Mongolia, from 2013 to early 2020, particularly during the resurgences in the Philippines in 2013 to 2014 and in 2017 to 2019. Genotype B3 continued to be reported by Malaysia from 2022 to 2023 and was detected as imported in Australia, Singapore, New Zealand and China in 2023.

Genotype D8, which was considered endemic in Malaysia since 2015 and in Viet Nam in 2017–2020, continued to be detected as imported in all countries and areas of the region verified to have achieved measles elimination, except Mongolia, from 2013 to early 2020. Genotype D8 continued to be reported by Malaysia from 2022 to 2023 and as imported in several countries and areas of the region which were verified to have achieved measles elimination during this same period.

Since 2000, ten different measles virus genotypes have been detected in the WPR (B3, D3, D4, D5, D8, D9, D11, G3, H1 and H2). Currently only two genotypes, B3 and D8, continue to be detected and, although none of the remaining eight genotypes have been detected in the last four years, it is interesting to note that D5 “disappeared” in 2008 from China and Japan only to re-appear briefly in China in 2015, which has implications for when to consider a genotype as being finally eliminated.

### 3.6. Prevention of Endemic Measles Virus Transmission Resurgence

The increased transmission of the genotype H1 measles virus was one of major causes of the regional resurgence of measles in the Western Pacific from 2013 to 2016 but has steadily declined in China from 2014 to 2019 and not been detected since September 2019 either by China or any other country or area in the region. This reflects China’s commitment to achieve a very high two-dose routine MCV coverage in all sub-populations and for provinces to implement targeted annual SIAs in lower coverage areas. With the delayed nation-wide MCV-SIA conducted in the Philippines in 2020, the resurgence of the genotype B3 measles virus in 2018 to 2019 could not be prevented in the country and there was the international seeding of virus again. The endemic measles transmission of genotype D8 continues in Malaysia with cyclic resurgences in 2012, 2015–2018 and 2023 onwards, with specific sub-populations continuing to miss out on routine immunizations.

### 3.7. Prevention of Large-Scale Import-Related Measles Outbreaks

During the 2018–2019 global measles resurgence, large-scale outbreaks were prevented after importation of measles cases in several countries in the WPR (e.g., Australia, Brunei Darussalam, Japan and the Republic of Korea). Hong Kong SAR (China), Macao SAR (China) and Singapore experienced import-related measles outbreaks, but their elimination status was maintained due to excellent outbreak response and existing high population immunity. New Zealand was able to maintain its elimination status despite experiencing a large-scale import-related measles outbreak from January 2019 to July 2020 [28] but remains under threat for resurgences due to falling immunization rates, particularly in the Māori and Pacific population.

Cambodia experienced multiple outbreaks in 2019 to 2020 due to measles virus importations from neighbouring countries with the increased endemic transmission of genotype D8 measles virus in 2018–2019. This resulted in the loss of measles elimination status for the country, which had been verified in 2015. Samoa and Tonga experienced large-scale import-related measles outbreaks from 2019 to 2020 [28] but the Tongan outbreak was rapidly extinguished due to prevailing high routine MCV coverage.

### 3.8. Interruption of Ongoing Transmission of Endemic Measles Virus

The genotype D9 measles virus, which was endemic in Malaysia and reported as imported from some countries in the region where measles elimination was verified, has not been detected by either Malaysia or any other country since January 2018.

Genotype H1 measles virus, which was endemic in and reported by China until the third quarter of 2019 and detected as imported by countries verified to have achieved measles elimination, has not been detected by either China or any other country in the region since September 2019.

Mongolia and Cambodia, which the RVC concluded in 2016 and 2021, respectively, had re-established endemic measles virus transmission, and the Pacific sub-region, in which Samoa and Tonga experienced import-related large-scale measles outbreaks in 2019, have achieved and sustained very low measles incidence (<1 per one million population) for at least three years (since 2018 in Mongolia and since 2021 in Cambodia and the Pacific sub-region).

### 3.9. Rubella Elimination

Since the verification process for rubella elimination started in 2017, seven countries and areas in the WPR (the Republic of Korea, Australia, Hong Kong SAR (China), Singapore, New Zealand, Macao SAR (China) and Brunei Darussalam) have been verified to have achieved and sustained rubella elimination as of September 2023. Malaysia, Mongolia and the Pacific sub-region have established 2025 as their target year for rubella elimination. But several countries (e.g., Cambodia, China, Lao PDR, PNG, and the Philippines) are yet to establish an elimination target year.

In recent years, a higher proportion of rubella cases occurred among people > 14 years of age in the WPR. China experienced a resurgence of rubella virus transmission in 2018 to 2019, with most cases being individuals aged 15 to 25 years, reflecting immunity gaps among birth cohorts that were born prior to and around the time of rubella-containing vaccine introduction into the routine immunization schedule [33]. Japan had large outbreaks of rubella during 2012–2013 and 2018–2019 with most cases occurring among unvaccinated male adults, reflecting immunity gaps among the male birth cohorts from 1962 to 1979 that were historically not targeted for rubella-containing vaccine administration in routine immunization. A one-dose catch-up vaccination campaign was launched in 2019, targeting male birth cohorts [34]. The Philippines have consistently reported a high number of rubella cases over the past decade. There were large-scale rubella outbreaks in 2017, 2019 and 2020 and a significant proportion of rubella cases were reported among women of child-bearing age. The vaccination status for these groups was either undocumented or not vaccinated [35].

Several countries in the region, including China and the Philippines, have not yet established nation-wide CRS surveillance.

### 3.10. Expanding Laboratory Capacity in the Western Pacific Region

The measles and rubella laboratory network in the WPR has played a very significant role in its successes in eliminating these diseases. As other countries move towards elimination, this role will increase, particularly molecular detection and genotyping. During nation-wide measles resurgences in the Philippines, a single national laboratory for VPDs could not manage the huge increase in sample testing. To address this challenge, seven WHO accredited sub-national (referral) laboratories were established in the country. The establishment of more national VPD laboratories in countries of the Pacific Islands, which require in-country testing due to their remoteness, would greatly decrease the turnaround times inherent in the referral system. Six countries (Cook Islands, Kiribati, Solomon Islands, Tonga, Vanuatu and Western Samoa) have been identified for expansion and training is underway in their laboratories. Testing is intended to be operational by 2025.

The implementation of molecular detection methods is required in Cambodia, Lao PDR, Mongolia and Papua New Guinea. Therefore, Labshot 2030 was devised as a two-stage strategy: an assessment of priority countries (Cambodia, Fiji, Lao PDR, Malaysia, Mongolia, Papua New Guinea, Philippines and Viet Nam) and a road map drafted to integrate surveillance expertise and infrastructure leading to independent and sustainable laboratories within these countries. Reverse transcription polymerase chain reaction (RT-PCR) is important for detecting the measles or rubella viruses. However, it is possible to detect the vaccine strain (genotype A) post vaccination by this method, too. Epidemiological data are essential to suggest whether detection is vaccine-related, and confirmation is possible by vaccine-specific RT-PCR or genotyping. Once genotyping is established, it will provide faster results and more effective surveillance.

## 4. Conclusions

Notable progress towards measles and rubella elimination has been made in the WPR in the last 10 years, despite the region-wide measles resurgence in 2013–2016, the global measles resurgence in 2018–2019 and the COVID-19 pandemic in 2020–2023. This has provided empiric evidence that: (1) measles and rubella elimination are not only achievable but also sustainable even in countries with large populations (e.g., Japan and the Republic of Korea) under the pressure of multiple importations of measles virus from countries with endemic transmission or large-scale import-related outbreaks during region-wide and world-wide resurgences of measles virus transmission; (2) the interruption of endemic measles virus, including in one of the most populous countries in the world (China), appears feasible, as demonstrated through strong epidemiologic and virologic surveillance systems within the country and evidence throughout the WPR [36]; (3) the prevention of resurgence and large-scale import-related outbreaks is achieved where strong routine immunization programmes are maintained for extended periods (at least 10 years) with two doses of MRCV and high vaccination coverage (at least 90% for both with reliable coverage denominators), even in countries with large populations (e.g., China and Japan); and (4) many countries and areas in the WPR have established strong and resilient immunization programmes, including sensitive case-based surveillance systems with the strong support of public health laboratories across the region, that demonstrated sustained high routine immunization coverage during the COVID-19 pandemic.

This steady progress and considerable achievement have been made possible by the political commitments of the governments of the WHO’s Member States and strong advocacy by the WHO’s Regional Committee. The intensive sharing of lessons learned by countries’ National Immunization Programmes during regular regional meetings organized by WHO has resulted in the translation of success across vast geographic regions and populations. Technical recommendations by regional immunization experts through the annual Technical Advisory Group (TAG) for Immunization and Vaccine-Preventable Diseases in the Western Pacific Region and the annual review and recommendations provided by the Regional Verification Commission for Measles and Rubella Elimination in the Western Pacific to each National or Sub-Regional Verification Committee for Measles and Rubella Elimination have been instrumental in bolstering these achievements. The proactive updating and active implementation of the regional and national operational targets and strategies for elimination with full consideration of country-specific situations and local disease epidemiology is indispensable for accelerating progress and sustaining gains at both national and regional levels.

Major challenges to achieving and sustaining measles and rubella elimination are: (1) the resurgence of virus transmission still occurring in many countries globally with the increased risk of virus importation into countries which have already achieved elimination or are approaching elimination; (2) weak routine immunization programmes (e.g., low vaccination coverage not yet reached >80%, target population underestimated or not included in the coverage denominator, etc.) in a few priority countries; (3) the delayed implementation of proactive and preventive supplementary immunization activities to fill residual or emerging immunity gaps before the accumulation of susceptibles that are found by imported virus; and (4) a growing proportion of adult populations susceptible to rubella infection and transmission in countries with large populations.

To overcome those challenges, a stronger global commitment to coordinating and synchronizing elimination efforts and progress across the world is essential. Countries should continue strengthening their health systems, particularly their immunization programmes. Not addressing, detecting and filling immunity gaps with the transmission of vaccine-preventable diseases among older age groups (i.e., adolescents and young adults) and high-risk groups with low vaccination coverage (e.g., ethnic minorities, migrants) will continue to threaten the achievement and maintenance of elimination.

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
