# Peer review of "Measles and Rubella Elimination in the Western Pacific Region in 2013–2022: Lessons Learned from Progress and Achievements Made during Regional and Global Measles Resurgences"

_vaccines, 2024, doi:10.3390/vaccines12070817_

Round 1
Reviewer 1 Report
Comments and Suggestions for Authors
I think that this is a valuable report to understand progress and present situation on measles and rubella elimination activities in the WPR. I would like to commend the authors for their efforts in summarizing the complex situations across many countries in the Region.
My suggestions are as follows;
# Could you describe what countries that have achieved measles and rubella elimination have in common in order to achieve elimination? In other words, could you suggest what should be done in countries in common that have not achieved elimination?
Also, could you describe what countries that have not achieved elimination have in common? In other words, could you suggest what improvements in common should be made immediately in countries that have not achieved elimination?
# WPR achieved polio eradication earlier following PAHO. Could you describe what technical differences between polio eradication activity and measles-rubella elimination activity and what WPRO as the secretariat feels about them?
# Genotyping should be even more important in the future, and it is very appropriate to expand laboratory capacity in the Region. I also recommend adding that the laboratory network in the Region has played a major role so far.
# I also recommend adding the existence of type A for measles genotypes and its meaning to the genotype description.
# It goes without saying that the two-doses policy has contributed to a sharp decline in the incidence of measles and rubella, but the timing of vaccination is not necessarily standardized within the Region. How does WPRO evaluate this?
# On the conclusion, paragraph l.359, it states the following:
(4) Many countries and areas in the WPR have established strong and resilient immunization programmes, including sensitive case-based surveillance systems with strong support of public health laboratories across the Region, that demonstrated sustained high routine immunization coverage during the COVID-19 pandemic.
During the COVID-19 pandemic, routine immunization coverage has declined in many countries in the world, and outbreaks of VPD such as measles, diphtheria, whooping cough, etc. have been reported, or there are fears of a resurgence of VPDs. I have heard the same situation in the WPR, but if this is the case, then the statement in (4) seems incorrect.
Author Response
Comment or Suggestion from Reviewer 1-1: Could you describe what countries that have achieved measles and rubella elimination have in common in order to achieve elimination? In other words, could you suggest what should be done in countries in common that have not achieved elimination?
Response from Authors 1-1: ”(3) prevention of resurgence and large-scale import-related outbreaks is achieved where strong routine immunization programmes are maintained for extended periods (at least 10 years) with 2 doses of MRCV and high vaccination coverage (at least 90% for both with reliable coverage denominators), even in countries with large populations (e.g. China and Japan)” (Lines 368 to 372).
Comment or Suggestion from Reviewer 1-2: Also, could you describe what countries that have not achieved elimination have in common? In other words, could you suggest what improvements in common should be made immediately in countries that have not achieved elimination?
Response from Authors 1-2: “Major challenges to achieving and sustaining measles and rubella elimination are: .. .. (2) weak routine immunization programmes (e.g. low vaccination coverage not yet reached >80%, target population underestimated or not included in the coverage denominator, etc.) in a few priority countries; (3) delayed implementation of proactive and preventive supplementary immunization activities to fill residual or emerging immunity gaps before accumulation of susceptibles that are found by imported virus” (Lines 392 to 399).
“Countries should continue strengthening their health systems, particularly their immunization programmes. Not addressing, detecting and filling immunity gaps with transmission of vaccine-preventable diseases among older age groups (i.e. adolescents and young adults) and high-risk groups with low vaccination coverage (e.g. ethnic minorities, migrants) will continue to threaten the achievement and maintenance of elimination” (Lines 403 to 408).
Comment or Suggestion from Reviewer 2: WPR achieved polio eradication earlier following PAHO. Could you describe what technical differences between polio eradication activity and measles-rubella elimination activity and what WPRO as the secretariat feels about them?
Response from Authors 2: Major differences between polio, polio virus and polio immunization and measles, measles virus and measles immunization are: (1) while approximately 90% of all infections of polio virus are inapparent or result in nonspecific fever, almost all infections of measles virus among susceptible people are apparent with typical symptoms e.g. fever, dry cough, runny nose, conjunctivitis and rash; (2) while the basic reproductive number of polio is 5-7, the one of measles is 12-18; and (3) while OPV that has been used for polio eradication can spread from vaccinees to surrounding susceptible individuals through fecal-oral route and immune them, live attenuated measles containing vaccine that has been used for measles elimination can immune only vaccinees. These differences between polio and measles have resulted that geographic and demographic distribution of measles virus transmission has been more easily and strategically described country by country and within countries as well more specifically, enabling to develop country-specific immunization strategies (e.g. whether or not, and when, if needed, mass vaccination campaign should be carried out to prevent resurgence of endemic virus or large-scale outbreaks of imported virus of measles, where or in which specific groups susceptible populations more quicky accumulates, which require specific vaccination intervention at sub-national levels, etc.). Per changing epidemiology of measles between and within countries, assessing susceptibility profile among populations and risk of resurgence or import-related outbreaks should be regularly carried out and country-specific intervention e.g. immunization strategies, outbreak preparedness, etc. should be carried out for achievement and maintenance of measles elimination.
Comment or Suggestion from Reviewer 3: Genotyping should be even more important in the future, and it is very appropriate to expand laboratory capacity in the Region. I also recommend adding that the laboratory network in the Region has played a major role so far.
Response from Authors 3: We have added one sentence (Lines 335-337)
Comment or Suggestion from Reviewer 4: I also recommend adding the existence of type A for measles genotypes and its meaning to the genotype description.
Response from Authors 4: We have added a few sentences (Lines 351-356)
Comment or Suggestion from Reviewer 5: It goes without saying that the two-doses policy has contributed to a sharp decline in the incidence of measles and rubella, but the timing of vaccination is not necessarily standardized within the Region. How does WPRO evaluate this?
Response from Authors 5: While all countries in the Region except one small Pacific country (i.e. Vanuatu) completed introduction of the second dose of measles containing vaccine into the routine childhood immunization programme before 2010, just “two-doses policy” did not result in “a sharp decline in the incidence of measles and rubella”. “A sharp decline in the incidence of measles and rubella” has often occurred right after implementation of high-quality nation-wide mass vaccination campaign. As the size of population, age-distribution of susceptibles, coverage of routine vaccination and immunization strategies varies country by country in the Western Pacific Region, the timing of vaccination and immunization strategy have had to be specific country by country and even within subnational level per specific measles epidemiology.
Comment or Suggestion from Reviewer 6: On the conclusion, paragraph l.359, it states the following: (4) Many countries and areas in the WPR have established strong and resilient immunization programmes, including sensitive case-based surveillance systems with strong support of public health laboratories across the Region, that demonstrated sustained high routine immunization coverage during the COVID-19 pandemic. During the COVID-19 pandemic, routine immunization coverage has declined in many countries in the world, and outbreaks of VPD such as measles, diphtheria, whooping cough, etc. have been reported, or there are fears of a resurgence of VPDs. I have heard the same situation in the WPR, but if this is the case, then the statement in (4) seems incorrect.
Response from Authors 6: “All countries and areas, which were verified by the RVC to have achieved and sustained measles elimination, maintained >90% annual coverage of measles and rubella containing vaccine first dose (MRCV-1) from 2013 to 2022, .. .. (Table 3). Of these countries, only New Zealand experienced notable declines in MRCV-2 coverage during COVID-19 pandemic in 2021-2022. Among countries which have not yet been verified to have achieved measles elimination, China and Malaysia sustained >90% annual coverage with both MRCV-1 and MRCV-2 in each year from 2013 to 2022 (except MRCV-2 coverage in 2019 (87.0%) and 2020 (84.0%) in Malaysia). Viet Nam, which reported >90% coverage of both MRCV-1 and MRCV-2 in each year from 2014 to 2020, experienced a marked decline in coverage with both MRCV-1 and MRCV-2 during the COVID-19 pandemic response in 2021 and 2022. Routine coverage of both MRCV-1 and MRCV-2 never reached 80% in the Philippines and Papua New Guinea from 2013 to 2022, and the Philippines had decreasing coverage with both MRCV-1 and MRCV-2 during the COVID-19 pandemic response in 2021 and 2022” (Lines 171-186). Therefore, only few countries in the Western Pacific Region (e.g. New Zealand, Viet Nama and the Philippines) experienced declined routine immunization coverage during the COVID-19 pandemic. The above-mentioned situation (i.e. “routine immunization coverage has declined in many countries in the world, and outbreaks of VPD such as measles, diphtheria, whooping cough, etc. have been reported, or there are fears of a resurgence of VPDs”) did not occur in the Western Pacific Region in 2020-2022 (and 2023 as well).

Reviewer 2 Report
Comments and Suggestions for Authors
This review by Takashima nicely covers the status of measles and rubella cases and outbreaks in the Western Pacific Region from the last two decades. It discusses vaccine coverage rates as well as the viruses that were detected and how the countries have been able to target vaccination campaigns to decrease cases. The paper is well written and covers an interesting topic of vaccination coverage within a region. I only had one minor comment that would help the reader determine the region covered early on in the paper. It would be useful to define the countries within the WPR early in the paper. Could either be in the text or by using a map, but within the first paragraph would be nice.
Table 4 was fuzzy and difficult to read, please use different formatting to enable readability.
Table 6 was very small making it difficult to read
Comments on the Quality of English LanguageThe second sentence of the abstract is a little awkward and could be reworded, but other than that, nicely written
Author Response
Comment or Suggestion from Reviewer 1: I only had one minor comment that would help the reader determine the region covered early on in the paper. It would be useful to define the countries within the WPR early in the paper. Could either be in the text or by using a map, but within the first paragraph would be nice.
Response from Authors 1: We have added Figure 2, showing maps of the Region (considering the structure and flow of the manuscript, the maps have been placed in Section 3, not in Section 1)
Comment or Suggestion from Reviewer 2: Table 4 was fuzzy and difficult to read, please use different formatting to enable readability.
Response from Authors 2: We’ve revised the Table and tried to make this Table less fuzzy using larger fonts and highlighted important data.
Comment or Suggestion from Reviewer 3: Table 6 was very small making it difficult to read.
Response from Authors 3: We’ve revised the Table and tried to make this Table a bit more readable.

Reviewer 3 Report
Comments and Suggestions for Authors
Manuscript type: Review
Title: Good
· L 54 – 56: In 2012, the Region recorded the lowest number of measles cases in its history – 10,794 cases, or 5.9 cases per 1 million population, compared with 177,052 cases, or 105.1 cases per 1 million population in 2000 (Figure 1) === What about years 2020, 2021, and 2022. I think they better than records in 2012.
· L 87 – 90: Malaysia, where genotype D8 measles virus has continued to be detected since 2013 and genotype D9 virus in 2015-2017, experienced increased measles virus transmission between 2015 and 2018 === provide a suitable reference.
· L 185 – 187: Japan, Republic of Korea, Australia, Hong Kong SAR (China), Singapore, New Zealand, Macao SAR (China) and Brunei Darussalam were verified by the RVC to have achieved and then verified to have sustained measles elimination without implementation of nation-wide MCV-SIAs between 2013 and 2023 === how you can interpret it?
· L 193: Mongolia predominantly affected children aged 0 to 8 months === rewording
References:
· Ref 4 – 8, 18, 20 – 24, 28, 29, 31: add the relevant links
Comments on the Quality of English LanguageFine
Author Response
Comment or Suggestion from Reviewer 1: L 54 – 56: In 2012, the Region recorded the lowest number of measles cases in its history – 10,794 cases, or 5.9 cases per 1 million population, compared with 177,052 cases, or 105.1 cases per 1 million population in 2000 (Figure 1) === What about years 2020, 2021, and 2022. I think they better than records in 2012.
Response from Authors 1: We have described here measles situation in the Western Pacific Region from 2000 to 2012 as Background (Section 1), which is then followed by Section 2 on the 2013-16 Region-wide measles resurgence, and progress made thereafter (2017 to 2022) (Section 3). Comments on low measles incidence in 2019-22 are stated in section 3.2.
Comment or Suggestion from Reviewer 2: L 87 – 90: Malaysia, where genotype D8 measles virus has continued to be detected since 2013 and genotype D9 virus in 2015-2017, experienced increased measles virus transmission between 2015 and 2018 === provide a suitable reference.
Response from Authors 2: Please refer to Table 5, which shows that Malaysia detected genotype D8 measles virus in 2013, 2015-2019 and 2022-2023 and genotype D9 measles virus in 2015-2017 and Table 2, where Incidence of Measles (per million population) in Malaysia continued to increase from 42.4 (2015), 50.3 (2016), 52.9 (2017) to 60.4 (2018).
Comment or Suggestion from Reviewer 3: L 185 – 187: Japan, Republic of Korea, Australia, Hong Kong SAR (China), Singapore, New Zealand, Macao SAR (China) and Brunei Darussalam were verified by the RVC to have achieved and then verified to have sustained measles elimination without implementation of nation-wide MCV-SIAs between 2013 and 2023 === how you can interpret it?
Response from Authors 3: “(3) prevention of resurgence and large-scale import-related outbreaks is achieved where strong routine immunization programmes are maintained for extended periods (at least 10 years) with 2 doses of MRCV and high vaccination coverage (at least 90% for both with reliable coverage denominators), even in countries with large populations (e.g. China and Japan)” (Lines 368 to 372)
Comment or Suggestion from Reviewer 4: L 193: Mongolia predominantly affected children aged 0 to 8 months === rewording
Response from Authors 4: We’re reworded (Lines 199-201).
Comment or Suggestion from Reviewer 5: Ref 4 – 8, 18, 20 – 24, 28, 29, 31: add the relevant links
Response from Authors 5: Done.
